# ScaleSim: Serving Large-Scale Multi-Agent Simulation with Invocation Distance-Based Memory Management

Zaifeng Pan [1] [*]  Yipeng Shen [1] [*]  Zhengding Hu [1]  Zhuang Wang [2]  Aninda Manocha [2]  Zheng Wang [1]
Zhongkai Yu [1]  Yue Guan [1]  Yufei Ding [1]

## Abstract

LLM-based multi-agent simulations are increasingly adopted across application domains, but remain difficult to scale due to GPU memory pressure. Each agent maintains private GPU-resident states, including models, prefix caches, and adapters, which quickly exhaust device memory as the agent count grows. We identify two key properties of these workloads: sparse agent activation and estimable agent invocation orders. Based on an analysis of representative workload classes, we introduce invocation distance, a unified abstraction that estimates the relative order in which agents will issue future LLM requests. Leveraging this abstraction, we present ScaleSim, a memory-efficient LLM serving system for large-scale multi-agent simulations. ScaleSim enables proactive prefetching and priority-based eviction, supports diverse agent-specific memory through a modular interface, and achieves up to 1.74× speedup over SGLang on simulation benchmarks. ScaleSim's source code is available at https://github.com/PanZaifeng/KVFlow.

## 1. Introduction

Large language model (LLM)-based multi-agent simulation is widely used to model and analyze complex systems involving interacting agents. It has enabled progress in diverse application domains, including social science (Park et al., 2023; 2024; Ren et al., 2024; Li et al., 2025b), autonomous driving (Gulino et al., 2023; Zhou et al., 2024), urban planning (Ni et al., 2024), robotics (Kannan et al., 2024; Ren et al., 2025), and economic modeling (Li et al., 2023). By capturing how individual agents perceive, decide, and act

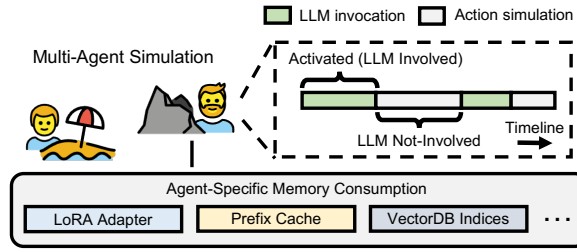

*Figure 1.* Illustration of multi-agent simulation phases and agent-specific memory consumption.

in shared environments, multi-agent simulation serves as a key tool for studying collective behaviors, coordination strategies, and emergent dynamics in large-scale systems.

Scaling up the number of agents in simulation improves behavioral diversity and enables more complex interaction patterns, which are critical for many downstream tasks. However, existing backend systems for serving LLM-based agent applications are struggling to handle large-scale simulations efficiently. A key bottleneck arises from the *agent-specific memory*, which includes private models (Park et al., 2025), LoRA adapters (Yu et al., 2024), prefix caches (Zheng et al., 2024a; Gim et al., 2024; Pan et al., 2025b), and action histories[1] (Packer et al., 2023; Chhikara et al., 2025; Xu et al., 2025b). As illustrated in Figure 1, each agent maintains its own memory components on the GPU. As the agent population increases, the aggregate memory demand grows rapidly, often exceeding available GPU capacity. This results in frequent memory evictions and data transfers between host and device, introducing significant I/O overhead (Figure 2) and degrading overall system throughput.

Despite this growing memory footprint, we make a key observation: agent execution is inherently sparse. Specifically, the number of concurrent LLM requests at any given time is much smaller than the total number of agents. This sparsity arises from the two-phase structure of agent simulation shown in Figure 1: one phase involves invoking the LLM

---

[*]Equal contribution  [1]University of California, San Diego  [2]Amazon Web Services. Correspondence to: Zaifeng Pan <zapan@ucsd.edu>.

*Proceedings of the 43rd International Conference on Machine Learning*, Seoul, South Korea. PMLR 306, 2026. Copyright 2026 by the author(s).

---

[1]To avoid ambiguity, we use *action history* to represent the agent's accumulated experience or memory content, as *memory* in this paper refers to storage resources such as GPU memory.

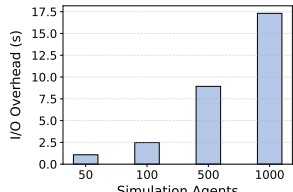

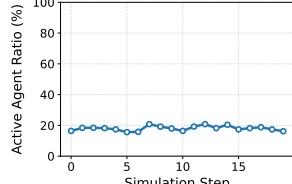

*Figure 2.* I/O overhead per simulation step increases with the number of simulation agents.

*Figure 3.* Only a small fraction of agents are activated during each simulation step.

to generate actions (LLM-involved), while the other executes those actions in the environment without requiring LLM inference (LLM-not-involved). We refer to an agent as *activated* when it is in the LLM-involved phase. Our profiling of large-scale agent societies (Piao et al., 2025) reveals that only a small fraction of agents are activated at any moment, resulting in sparse LLM execution patterns shown in Figure 3. This execution sparsity implies that simply adding more machines to handle scalability often leads to poor hardware utilization, since most agents are inactive at any given time.

We make another key observation: in many multi-agent simulation workloads, the relative order in which agents are invoked can be effectively estimated. Based on how agent invocation order can be estimated, we categorize representative applications into three types: (1) *Independent simulation*, where agents operate in isolation and the next invocation can be inferred directly from their current action; (2) *Interaction-involved simulation*, where agent interactions are possible, but future activations remain predictable based on environmental cues such as agent location and moving velocity; and (3) *Predefined activation paths*, such as information diffusion in social networks, where the next activation time of agents can be compared through structural patterns like hop counts.

Motivated by sparse agent activation and estimable invocation orders, we propose ScaleSim, an LLM serving system that efficiently manages GPU memory for large-scale multi-agent simulations. ScaleSim introduces *invocation distance* for each agent to guide memory management and exposes a unified interface for applications to provide distance estimates. Based on the invocation distances of agents, ScaleSim enables distance-aware eviction, proactive prefetching, and execution preemption, which substantially reduces I/O overhead. ScaleSim further defines an extensible abstraction for agent-specific memory, allowing heterogeneous memory modules to be seamlessly integrated and jointly managed. We implement ScaleSim on top of SGLang and evaluate it on three representative simulation applications: AgentSociety (Piao et al., 2025), Generative Agents (Park et al., 2023), and information diffusion (Gao et al., 2023). Experimental results demonstrate

that ScaleSim provides consistent performance improvements as the number of agents scales.

In summary, we make the following contributions:

- We identify two key properties of multi-agent simulation workloads: sparse agent activation and estimable agent invocation orders. To capture these behaviors, we introduce the unified invocation distance abstraction, which generalizes activation patterns across diverse applications.

- We design and implement ScaleSim, a memory-efficient LLM serving system that leverages invocation distance to guide memory management decisions.

- We conduct comprehensive experiments on three representative workloads. Results show that ScaleSim achieves up to $1.74\times$ speedup compared to a vanilla SGLang baseline, demonstrating its effectiveness in scaling multi-agent simulation.

## 2. Background and Related Work

**LLM-based multi-agent simulation.** LLM-based multi-agent simulation models a system composed of multiple autonomous agents, where each agent relies on a large language model to reason about its local state and generate actions in response to the shared environment. As illustrated in Figure 1, the behavior of each LLM-based agent typically alternates between two phases: invoking the LLM to generate the next action, and executing that action over a period of time without LLM involvement. This results in the sparse activation patterns shown in Figure 3, where only a small subset of agents are simultaneously issuing LLM requests. This structural property introduces opportunities for more efficient memory and compute management in large-scale systems.

**Agent-specific memory.** Each agent in an LLM-based simulation maintains its own memory footprint, comprising both static model configurations and dynamic runtime state. On the static side, agents may load personalized components such as LoRA adapters (Yu et al., 2024), system-specific prefix caches that encode role and behavioral priors (Zheng et al., 2024a; Pan et al., 2024; Gim et al., 2024; Pan et al., 2025a;b), or even auxiliary draft models used for speculative decoding (Leviathan et al., 2023; Liu et al., 2023; Miao et al., 2024). These components are typically instantiated per agent and persist across multiple simulation steps. At runtime, agents accumulate action histories based on prior interactions (Packer et al., 2023; Ouyang et al., 2025), often requiring dedicated storage and indexing structures (Chhikara et al., 2025; Xu et al., 2025b) to support efficient retrieval. In retrieval-augmented generation

(RAG) scenarios (Lewis et al., 2020; Zhang et al., 2021; Xu et al., 2025a), agents further exhibit diverse and skewed access patterns depending on their task focus, resulting in heterogeneous caching demands across GPU and CPU memory (Hu et al., 2025; Lin et al., 2025; Jin et al., 2024). As illustrated in Figure 1, these components together define the agent-specific memory footprint, which must be efficiently managed to support large-scale simulation.

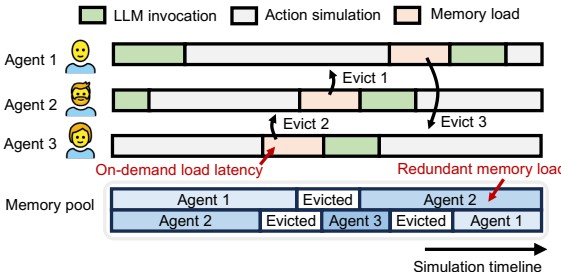

*Figure 4.* Inefficiency of existing systems due to frequent reactive memory swaps and suboptimal eviction decisions.

**LLM serving systems.** LLM-based multi-agent simulation applications rely on general-purpose LLM serving systems such as vLLM (Kwon et al., 2023) and SGLang (Zheng et al., 2024a) to provide backend inference for agent decision making. These systems incorporate optimizations including flexible batching (Yu et al., 2022; Agrawal et al., 2024; Zheng et al., 2024b; Chen et al., 2024b; Sheng et al., 2024; Guan et al., 2025), KV cache management (Qin et al., 2025; Liu et al., 2024; Sheng et al., 2023; Zhang et al., 2025a), large-scale parallelization (Patel et al., 2024; Zhong et al., 2024; Chen et al., 2024a; 2026), and kernel-level optimizations (Dao et al., 2022; Ye et al., 2025; Dong et al., 2024; Pan et al., 2025a). However, these systems are unaware of simulation semantics, leading to inefficient memory management in large-scale multi-agent simulation workloads. We detail their inefficiency in Section 3.1.

While recent works explore system optimizations for agentic workloads, they primarily target scenarios other than general multi-agent simulation or focus on more specialized settings. Workflow-driven systems (Lin et al., 2024; Pan et al., 2025b; Gim et al., 2025) assume pre-defined execution structures, making them unsuitable for dynamic, large-scale simulations with decentralized control. Infer-Cept (Abhyankar et al., 2024) manages KV cache in tool-augmented LLMs using profiled tool execution time, but does not account for multi-agent simulations where action execution time varies across agents and steps and may depend on inter-agent interactions. AI Metropolis (Xie et al., 2024) improves batch efficiency in a specific simulation setting. It assumes that action simulation does not affect subsequent LLM generations to perform out-of-order execution, which does not hold in many multi-agent simulation applications. We provide a more detailed comparison with prior systems in Appendix B.

In this paper, we focus on the memory efficiency of LLM serving systems for multi-agent simulation. Compared to existing works, our approach applies to a broader range of scenarios beyond their assumptions.

## 3. Invocation Distance-Based Memory Management

### 3.1. Limitation of Existing Systems

Existing LLM serving systems (Kwon et al., 2023; Zheng et al., 2024a; NVIDIA, 2025) are designed to be application-agnostic: they treat each request as an independent invocation and have no visibility into the execution semantics of the frontend application. As a result, these systems manage agent-specific memory using generic strategies such as Least Recently Used (LRU) or usage-based heuristics, without leveraging any information about when each agent will be activated again. While this approach works adequately for stateless inference workloads, it becomes a fundamental bottleneck in multi-agent simulation. We observe two key limitations of existing systems in this setting:

**Frequent reactive memory swaps.** As illustrated in Figure 4, consider a simulation involving three agents, where the available GPU memory can accommodate only two agents' memory at a time. In this setting, current LLM serving systems, which lack visibility into application-level scheduling, can only load an agent's memory on demand, that is, when the agent issues an LLM call. For example, at the beginning of the simulation, Agent 1 and Agent 2 each issue an LLM call. Since the system loads their corresponding memory to the GPU for inference, the available memory is fully occupied, so at this point, Agent 3's memory cannot be cached on the GPU. Later, when Agent 3 becomes active and attempts to perform an LLM inference, the system must reactively load its memory from CPU to GPU. This memory swap occurs despite the fact that both Agent 1 and Agent 2 are no longer in their LLM stages and have already transitioned to the simulation phase. However, because their memory remains on the GPU without being reclaimed, Agent 3 incurs a cold start cost.

Such on-demand memory swapping introduces repeated transfers between CPU and GPU memory, which are costly and delay the execution of LLM inference. Since each memory load incurs high latency, especially when model-specific

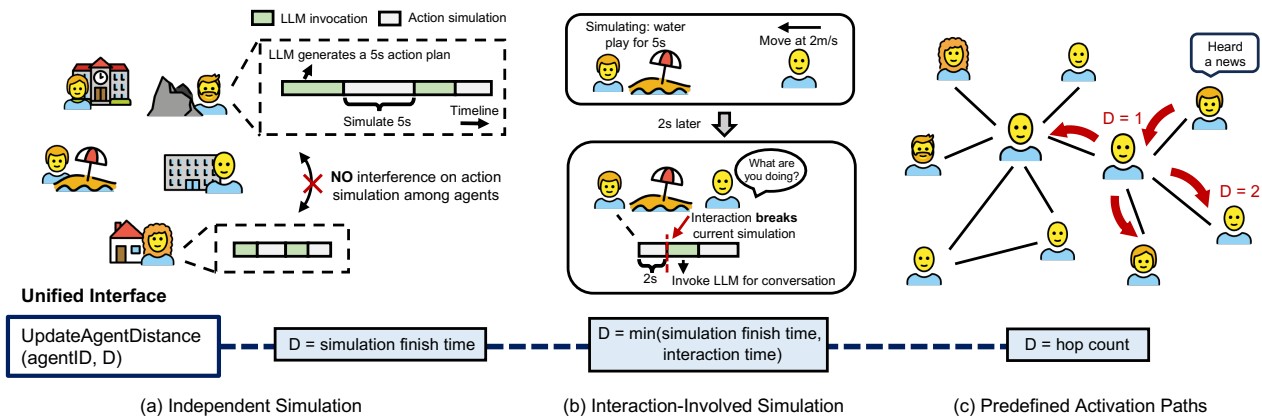

*Figure 5.* Three representative categories of multi-agent simulation workloads. In each case, we define the invocation distances of agents accordingly and pass them to ScaleSim through a unified interface.

memory such as LoRA adapters and prefix cache can be very large, these stalls accumulate and significantly degrade system throughput. Importantly, this inefficiency is not due to GPU compute saturation, but rather a consequence of treating agent activations as isolated events without considering their temporal relationships.

**Suboptimal eviction decisions.** Existing LLM serving systems make eviction decisions solely based on recent memory access history, without understanding the actual execution state of each agent. As a result, the system may incorrectly assume that an agent is inactive simply because it has not accessed the LLM for a while. In reality, the agent may just be in the middle of its simulation phase, during which it does not use the LLM but will soon issue another request. This misinterpretation often leads to premature eviction of memory that will be needed again shortly.

As illustrated in Figure 4, after Agent 2 completes its LLM generation, it proceeds to a local simulation phase where it no longer interacts with the LLM. During this period, the system mistakenly regards Agent 2 as a low-priority candidate for retention, since its memory has not been accessed recently. Consequently, when Agent 3 issues a new LLM request, the system evicts Agent 2's memory to free GPU space. However, Agent 2 is scheduled to resume LLM inference shortly afterward, forcing its memory to be reloaded soon. In contrast, Agent 1, which remains in a long simulation phase and will not use the LLM for an extended duration, retains its memory on the GPU unnecessarily.

This mismatch between recency-based eviction policies and the actual execution schedule of agents results in inefficient GPU memory usage. The system often reloads memory for agents that will soon become active, while unnecessarily retaining memory for agents that are in long simulation phases and will not invoke the LLM in the near future. The root cause is not the insufficiency of GPU memory itself, but

rather the system's lack of foresight into when each agent will next require model inference. Without such awareness, the memory manager cannot make informed decisions, leading to avoidable overhead and reduced throughput.

### 3.2. Invocation Distance Abstraction for Different Simulation Applications

To address the inefficiencies described above, we seek to bridge the gap between low-level memory management and high-level agent execution semantics. A key observation is that, in many multi-agent simulation applications, agent activations are not random but follow predictable patterns, providing an opportunity to anticipate when each agent will next issue an LLM request.

We leverage this insight by introducing a simulation-aware abstraction called *invocation distance*, which captures the temporal gap between an agent's current state and its next expected LLM invocation. Invocation distance does not necessarily represent an exact timestamp or number of steps; rather, it serves as a relative measure that enables the system to compare the urgency of different agents' upcoming activations. This allows memory management to move beyond reactive heuristics and instead make informed decisions by prioritizing agents with smaller invocation distances.

Based on how to construct the invocation distances, we classify multi-agent simulation applications into three representative categories:

**Independent simulation.** In independent multi-agent simulations (Wan et al., 2025; Wang et al., 2023a; Zhu et al., 2023; Zhou et al., 2023; Gur et al., 2023), each agent operates entirely in isolation, and its activation schedule is determined solely by its own internal logic. Since agents do not interact with each other, their execution timelines are decoupled and predictable. Figure 5 (a) illustrates an

example that each agent simulates independently, and the action simulation time can be directly inferred.

For instance, consider a scenario where a large number of robotic arms simultaneously perform repetitive tasks. Each arm repeatedly goes through a cycle consisting of observation, planning, and physical actuation. After completing an LLM-based planning step, the arm enters an action phase whose duration is fixed or bounded by system parameters. Similarly, in environments where multiple agents are independently playing through simulation episodes, each agent alternates between invoking the LLM to make decisions and executing rule-based or environment-driven actions. In both cases, the duration between two LLM calls can often be inferred from the agent's current phase.

In some applications (Piao et al., 2025), agents may exhibit limited forms of interaction, such as observing each other's behavior or logging information for future reference. However, these interactions do not interfere with the execution of the current simulation phase. They only affect the internal logs or memory buffers of the agents, which may influence future LLM generations. Since the ongoing simulation behavior remains unaffected, we categorize such settings as part of independent simulation.

In situations where this duration is not explicitly specified, the system can still estimate the invocation distance using application-specific rules or by prompting the LLM to predict a value based on the current context. These estimates do not need to be exact, as long as they capture the relative urgency among agents.

**Interaction-involved simulation.** Similar to independent simulations, each agent in this category (Park et al., 2023; 2024; Wang et al., 2023b; AL et al., 2024; Li et al., 2025a) follows a local execution loop where it generates an action plan through an LLM call and then performs the planned actions over a period of time. However, unlike in the independent case, agents may interact with others during this period, which can disrupt their current plan and trigger a new LLM invocation earlier than originally expected.

A representative example is Generative Agents (Park et al., 2023), where agents navigate a shared environment based on individually generated daily schedules. While agents typically simulate their plans independently, unexpected interactions can occur when they encounter each other in physical space. These interactions may lead to social conversations, collaborative task execution, or negotiation, each requiring a new LLM query to adjust the current course of action. As a result, the actual invocation time is influenced not only by the agent's own plan but also by potential interaction events.

To account for both sources of activation, we define the

invocation distance as the minimum of two components:

$$D = \min(D_{\text{action}}, D_{\text{interaction}}) \qquad (1)$$

where $D_{\text{action}}$ corresponds to the remaining duration of the current action sequence, and $D_{\text{interaction}}$ captures the expected time until the next possible interaction between agents. Generative agents and similar spatial multi-agent simulations constitute a specific example in this category, where the interaction distance can be estimated by computing the physical distance between agents and dividing by their movement velocity. As shown in Figure 5 (b), if two agents are moving toward each other in the environment, the time to interaction can be approximated as:

$$D_{\text{interaction}} = \frac{\text{Physical Distance}}{\text{Velocity}} \qquad (2)$$

This estimate enables the system to prioritize memory retention for agents that are spatially close to others and thus more likely to require a new LLM invocation in the near future, beyond only considering the current action duration.

**Predefined activation paths.** In some multi-agent simulations, agent activation is governed by structured propagation patterns that follow predefined paths through a network or environment (Gao et al., 2023; Li et al., 2024; Zhang et al., 2025b). These simulations are often used to model information diffusion, rumor spreading, or influence cascades in social networks (Gao et al., 2023). Unlike the previous categories, where agents actively perform or adjust actions based on internal plans or local interactions, agents in this category may remain inactive for extended periods until they are explicitly reactivated by upstream events.

A typical example is a simulation of information diffusion where activation proceeds in a breadth-first manner along a graph. An agent becomes active when it receives information from one of its neighbors, issues an LLM-based response, and then passes the information to others. After this step, the agent may become inactive for the rest of the simulation or remain idle until it receives new input. Although the agent performs no visible action during this waiting period, we treat it as a special form of action simulation, as it does not involve any LLM invocation.

In such settings, the system cannot rely on local execution state or physical proximity to estimate the next activation time. Instead, it can use the agent's position in the propagation path as a proxy for invocation distance. For example, in graph-based propagation, the hop count from the information source provides a natural and effective approximation of when an agent will next issue an LLM request, like Figure 5 (c) shows. Agents that are closer to the origin of propagation are likely to be activated earlier, whereas those farther away may remain idle for longer durations.

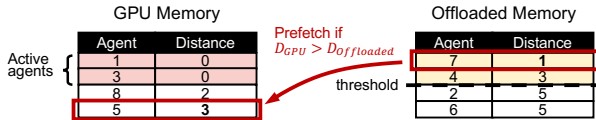

*Figure 6.* An example of invocation distance-guided proactive prefetching. Agent 7 is prefetched by replacing an inactive GPU-resident agent with a larger invocation distance.

### 3.3. Efficient Memory Management with Unified Interface

Building on the abstraction of invocation distance, we propose a memory management system, ScaleSim, that enables the backend LLM serving system to make informed and proactive decisions for multi-agent simulation workloads. This design bridges the semantic gap between the simulation frontend and the backend memory manager, allowing ScaleSim to coordinate memory usage more effectively under constrained GPU resources.

To support this coordination, ScaleSim provides a unified interface through which the simulation application exposes per-agent invocation distance estimates. Based on these values, we design two core optimizations in ScaleSim:

**Distance-guided proactive prefetch.** ScaleSim performs proactive prefetching by explicitly comparing invocation distances across agents. When an offloaded agent has an invocation distance below a predefined threshold, ScaleSim further checks whether its distance is smaller than that of any inactive agent currently resident in GPU memory. If sufficient GPU memory is available, either as free capacity or by replacing inactive agents with larger invocation distances, ScaleSim proactively prefetches the selected agent's memory from the CPU to the GPU. Since LLM inference does not involve CPU-GPU data transfer over PCIe, the prefetch operation can proceed in parallel without blocking the agent's execution on the GPU.

Figure 6 shows an example of this process. All active agents in GPU memory have an invocation distance of zero. Among offloaded agents, agents 7 and 4 satisfy the threshold condition. Agent 7 is prefetched because its invocation distance is smaller than those of inactive agents 8 and 5 on the GPU, whereas agent 4 is not prefetched since its invocation distance is larger than all inactive GPU-resident agents.

As shown in Figure 7, ScaleSim preloads Agent 3's memory while it is still performing actions. This allows the LLM call to begin immediately when triggered. In contrast, reactive memory loading causes the system to wait for the transfer to complete before inference can proceed. By overlapping memory loading with agent simulation, the proactive prefetching of ScaleSim significantly reduces I/O stalls and improves throughput.

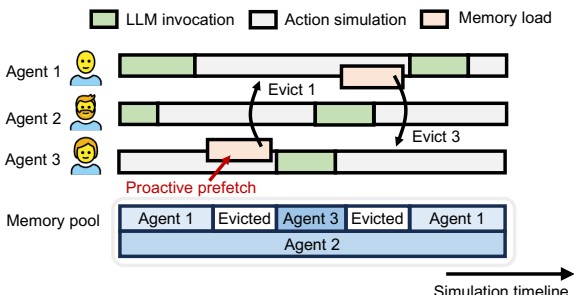

*Figure 7.* Simulation timeline after ScaleSim's optimization.

**Future reuse-aware eviction.** When GPU memory becomes insufficient, the system must evict memory belonging to one or more agents. Instead of relying on recency-based policies such as LRU, ScaleSim prioritizes eviction based on invocation distance to reduce redundant data transfers.

In Figure 7, when Agent 3 is about to issue a new LLM request, ScaleSim evicts Agent 1 rather than Agent 2, since Agent 2 has a shorter invocation distance and will be activated sooner. This decision avoids an additional memory load for Agent 2 and improves overall memory reuse.

This eviction strategy provides two benefits. First, even with prefetching, there are cases where memory transfers cannot fully overlap with computation, especially when many agents are prefetched in parallel. Reducing unnecessary memory loads at the source improves transfer efficiency. Second, for certain types of agent-specific memory such as prefix caches, users may choose not to back them up in CPU memory to save resources. Once evicted, such memory must be recomputed rather than restored. Avoiding eviction in these cases is important for maintaining both performance and resource efficiency.

## 4. Evaluation

We build the prototype of ScaleSim on top of SGLang (Zheng et al., 2024a) v0.5.2. To support heterogeneous agent-specific memory and enable unified, invocation distance–aware management in ScaleSim, we introduce an agent-specific memory abstraction class, which is detailed in Appendix A. Based on the interfaces defined by this abstraction, we implement two representative memory modules to demonstrate the applicability of our framework: one for managing LoRA adapters and another for handling prefix caches. In this section, we conduct the evaluation of ScaleSim to demonstrate its performance benefits in multi-agent simulation workloads.

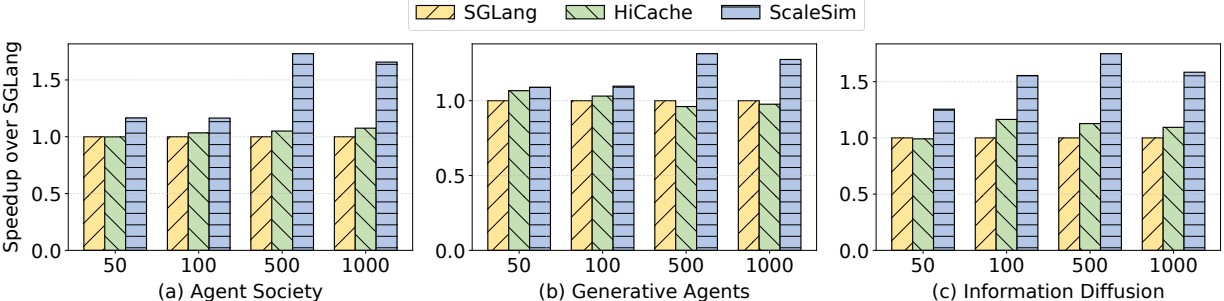

*Figure 8.* End-to-end speedup of ScaleSim over SGLang across different simulation applications and agent counts.

### 4.1. Experimental Setups

**Models and testbeds.** We evaluate ScaleSim on the Qwen2.5 series of instruction-tuned language models. We conduct most experiments on Qwen2.5-7B using an NVIDIA H100 GPU with 80GB memory and 64 GB/s PCIe Gen5 bandwidth. To evaluate ScaleSim's performance in multi-GPU settings, we further test on Qwen2.5-32B using up to 8 H100 GPUs interconnected via NVLink.

**Benchmarks.** To demonstrate the generality of ScaleSim across diverse multi-agent simulation workloads, we evaluate it on three applications, each representing a distinct invocation pattern: (1) For independent simulation, we use AgentSociety (Piao et al., 2025), an open-source simulator. Although agents in AgentSociety can have potential interactions, their actions are simulated independently without interference. (2) For interaction-involved simulation, we modify AgentSociety to add spatially driven interactions like Generative Agents (Park et al., 2023), where two agents initiate a conversation if their distance falls below a predefined threshold. (3) For predefined activation paths, we simulate the information diffusion over structured social networks according to a breadth-first schedule. The invocation distance is defined as the hop count from the network.

In all benchmarks, we assume each agent has its own LoRA adapter and prefix cache as the agent-specific memory. To simulate thousands of heterogeneous agents, we use a shared adapter for all agents but assign distinct logical IDs so the system treats them as separate.

**Baselines.** We compare ScaleSim with two configurations of SGLang (Zheng et al., 2024a) with S-LoRA support (Sheng et al., 2024). The first baseline, denoted as *SGLang*, uses a GPU-resident radix-structured prefix cache without CPU fallback. Once GPU memory is exhausted, evicted prefix nodes are discarded and must be recomputed upon reuse. The second baseline, *HiCache*, enables SGLang's default hierarchical caching, which asynchronously backs up frequently accessed prefix nodes to host memory. Upon reuse, these nodes can be restored

from CPU memory instead of being regenerated. Both baselines use an LRU policy for evicting agent-specific memory, including LoRA adapters and prefix caches, and perform memory loading reactively upon request.

### 4.2. End-to-End Performance

We measure the end-to-end speedup of ScaleSim over the baseline SGLang across three benchmark applications with varied simulated agent numbers. All experiments are conducted using Qwen2.5-7B-Instruct on a single H100 GPU. The active agent rate is around 20%, and to reflect GPU memory constraints, we cap the number of resident agents to 25% of the total (up to 125 on an H100 GPU).

Figure 8 reports the relative speedup of each method over the vanilla SGLang baseline. Across all benchmarks, ScaleSim consistently outperforms both SGLang and HiCache. In AgentSociety (Figure 8 (a)), where agents follow independent action simulation, ScaleSim achieves up to $1.73\times$ speedup by prefetching and pinning memory for agents with short invocation distances. In Generative Agents (Figure 8 (b)), the mobility-aware invocation distance allows ScaleSim to accurately prioritize memory for agents approaching interaction, yielding up to $1.31\times$ speedup. In information diffusion (Figure 8 (c)), the structured activation order enables ScaleSim to eliminate the cold-start of downstream nodes, reaching up to $1.74\times$ speedup.

Figure 9 shows the execution time per simulation step in AgentSociety as the number of simulated agents increases from 25 to 1,000. While all methods exhibit roughly linear growth as agent count increases, ScaleSim consistently incurs lower execution time compared to SGLang and SGLang w/ HiCache. This improvement stems from the efficient agent memory management of ScaleSim, which reduces the I/O overhead significantly. In contrast, the baselines suffer from higher memory churn due to their reactive eviction strategies, especially under high concurrency.

**Time-to-First-Token (TTFT) comparison.** TTFT is a critical latency metric for LLM-based multi-agent simula-

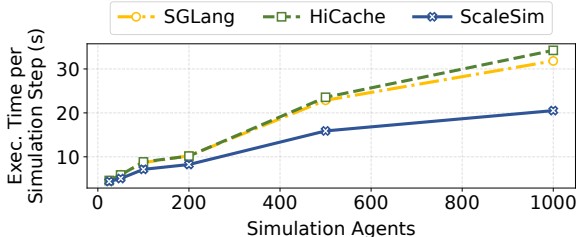

Figure 9. Per-step execution time scaling in AgentSociety.

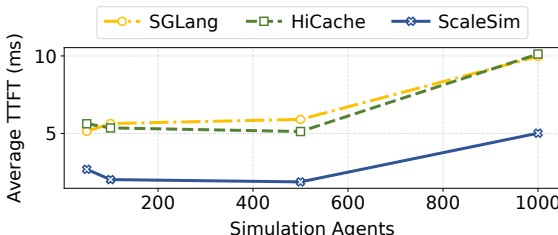

Figure 10. Average Time-to-First-Token (TTFT) in AgentSociety under increasing agent population.

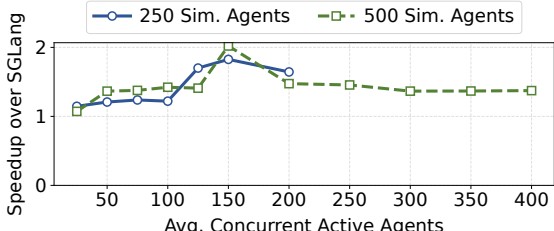

Figure 11. Speedup of ScaleSim under different levels of sparsity.

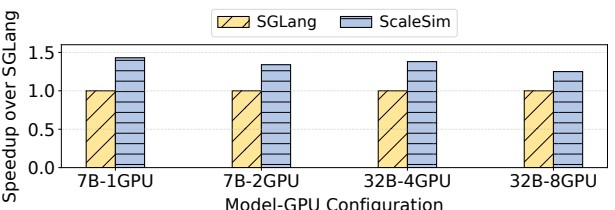

Figure 12. Speedup of ScaleSim across different model and GPU configurations with 500 simulated agents.

tions, especially in applications where outputs are streamed token-by-token to end users. A lower TTFT directly translates to better perceived responsiveness and service quality. Figure 10 reports the average TTFT across varying agent counts in the AgentSociety benchmark. As the number of simulated agents increases, both SGLang and HiCache experience significant TTFT degradation due to reactive memory loading and KV cache recomputation. In contrast, ScaleSim maintains substantially lower TTFT by proactively prefetching agent-specific memory with short invocation distances. Although TTFT increases with agent population for all methods, ScaleSim consistently reduces first-token latency by 48%-68% compared to HiCache under high agent concurrency, demonstrating its ability to improve not only throughput but also responsiveness.

**Sensitivity to sparsity.** Figure 11 evaluates the impact of workload sparsity on system performance under different levels of concurrency. We vary sparsity by controlling the simulation time of each frame in AgentSociety, which determines the degree of temporal overlap among agent executions. Results are reported for configurations with 250 and 500 total agents. For all these configurations, the GPU can host at most approximately 125 agents at the same time, and larger values on the x-axis correspond to denser execution overlap. Across a wide range of sparsity levels, ScaleSim consistently achieves meaningful speedups over the SGLang baseline, demonstrating its robustness beyond extremely sparse regimes.

**Sensitivity to prediction errors.** To evaluate robustness under inaccurate predictions, we conduct experiments by injecting random agent triggers at varying levels for the AgentSociety benchmark, thereby introducing controlled prediction errors. We report performance comparisons with SGLang for 500-agent simulations under different prediction error rates in Figure 13. Although the speedup decreases as the prediction error increases, ScaleSim still outperforms SGLang even under high error rates.

**Scalability across model and GPU configurations.** To evaluate the scalability of ScaleSim on AgentSociety under different hardware and model sizes, we run experiments using Qwen2.5 models of 7B and 32B parameters on 1, 2, 4, and 8 H100 GPUs. All multi-GPU setups use tensor parallelism, and the total number of simulated agents is fixed at 500. Figure 12 shows that ScaleSim consistently achieves higher speedup over SGLang across all configurations. This demonstrates that ScaleSim generalizes well to larger models and distributed inference without introducing performance bottlenecks.

**More diverse invocation distances.** We further evaluate ScaleSim on three extra benchmarks that exhibit diverse

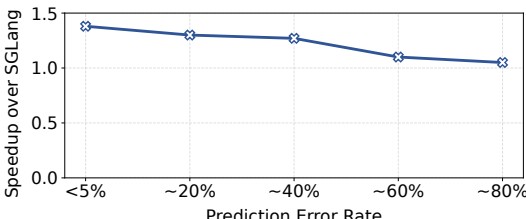

Figure 13. Speedup of ScaleSim over SGLang across agent across prediction error rates.

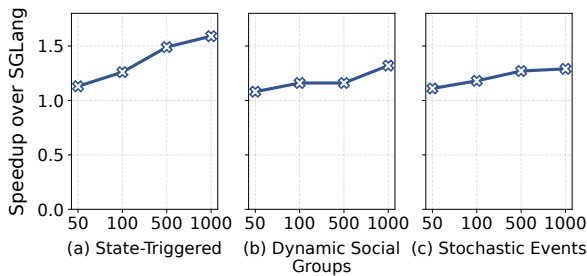

*Figure 14.* Speedup of ScaleSim over SGLang across agent counts on benchmarks with more diverse invocation distances.

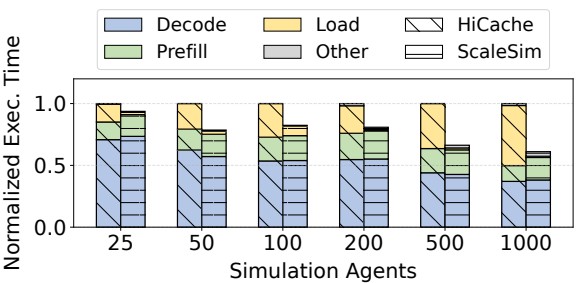

*Figure 15.* Execution time breakdown normalized to HiCache.

invocation distance patterns: (a) State-triggered simulation. Each agent maintains an internal state (e.g., "hunger level") that may interrupt its current action once a threshold is reached. We estimate the rate of state change based on recent observations to derive the invocation distance. (b) Dynamic social groups. In addition to physical distance constraints, agents interact only within their social groups, and these group memberships evolve over time. The invocation distance is thus defined based on dynamic social connectivity. (c) Stochastic environmental events. Beyond regular action execution, we introduce environmental events following a Poisson process, which may interrupt the simulation of a subset of agents. We incorporate the expected event trigger time when computing invocation distances. As shown in Figure 14, ScaleSim outperforms SGLang across these benchmarks, and the speedup increases as the number of the simulation agents grows.

### 4.3. Breakdown

Figure 15 presents a breakdown of normalized execution time for HiCache (left bars) and ScaleSim (right bars) under simulated agents varying from 25 to 1,000. Each bar is normalized to the total execution time of HiCache to highlight the relative contributions of different runtime components. We categorize the time into four parts: Load (host-to-GPU memory transfers), Prefill, Decode, and Other.

With ScaleSim, the time spent on Load is significantly reduced across all scenarios. This improvement is attributed to ScaleSim's proactive memory management, which prefetches high-priority agent memory before it is requested and overlaps it with the action simulation, avoiding blocking host-to-device transfers on critical paths. In contrast, HiCache relies on reactive memory loading, incurring repeated CPU-to-GPU transfers at runtime. ScaleSim does not directly affect the time spent on Prefill or Decode, which are dominated by model computation. As a result, the overall speedup is inherently bounded by the proportion of time spent in the Decode phase. Meanwhile, the Other category remains small and stable, indicating negligible CPU-side overhead and confirming that the invocation distance–aware

policy is lightweight and effectively hidden by overlap.

Meanwhile, the Other category remains small and relatively stable, indicating that ScaleSim does not introduce notable CPU-side scheduling overhead. This confirms that the invocation distance–aware policy is lightweight and can be further overlapped with other stages to hide latency, preserving both throughput and responsiveness.

## 5. Conclusion

In this work, we present ScaleSim, a system designed to address the inefficiencies of existing LLM serving frameworks when scaling to multi-agent simulations. We analyze representative classes of agent activation patterns and introduce the unified abstraction of *invocation distance* to guide efficient memory management decisions. Experimental results across multi-agent simulation benchmarks demonstrate the effectiveness and generality of ScaleSim.

## Acknowledgment

We sincerely thank the anonymous reviewers for their valuable feedback and insightful suggestions. This work is supported in part by UC AI LEAP and NSF grant 2607536.

## Impact Statement

This paper presents work whose goal is to advance the field of Machine Learning Systems. There are many potential societal consequences of our work, none which we feel must be specifically highlighted here.

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

# A. Agent-Specific Memory Abstraction

As multi-agent simulation applications continue to evolve, the types of memory required by each agent have become increasingly diverse, including prefix cache, LoRA adapters, draft model buffers, vector database indices, etc. Supporting these heterogeneous memory types poses new challenges for system design.

A naïve solution would be to reimplement memory management logic for each new memory type in an ad-hoc manner. However, this approach leads to duplicated engineering effort, increased code complexity, and difficulty in maintaining consistency across memory modules. Each new addition must separately handle eviction, prefetching, and registration logic, which increases the risk of bugs and performance regressions.

To improve extensibility and reduce implementation overhead, we provide an abstract base class for agent-specific memory in ScaleSim, which defines a set of common interfaces for memory management. Each memory module implements this interface and can be independently registered with the ScaleSim runtime. Once registered, the module is automatically integrated into ScaleSim's unified memory management workflow, including invocation distance-aware eviction and asynchronous prefetching.

Table 1 summarizes the major interfaces exposed by our abstraction. Each interface is invoked under a specific triggering condition, such as receiving an agent request, encountering memory pressure, or detecting an upcoming activation. This design allows the memory manager to coordinate agent-specific memory in a uniform and extensible manner.

*Table 1.* Invocation Distance-Based Memory Management Interfaces

| Memory Module Interface | Function | Triggered Condition |
|---|---|---|
| `HandleReq`(req, agentID) | Handle the LLM request from a specific agent and determine whether memory-related operations are required. This may involve establishing the agent–memory mapping, updating memory states, etc. | Triggered when the backend system receives an LLM request. |
| `Evict`(size, agentDistances) | Evict memory blocks based on invocation distance, prioritizing agents with larger values. Typically used to free up space for incoming activations. | Triggered when GPU memory is insufficient. Programmable by memory module developers. |
| `DispatchLoadTasks(` agentDistances, threshold, budget=None) | Selects a subset of agents whose invocation distance is below a threshold and asynchronously submits memory load tasks to the ScaleSim's scheduler. The scheduler will enqueue and process these tasks according to their priority and resource availability. | Triggered when the frontend sends a prefetch request. Programmable by frontend application developers. |
| `Load`(memObj, agentIDs, GetPreemptionSignal) | Load memory objects from lower-level storage (e.g., CPU). `GetPreemptionSignal` is a function handle that developers can query to determine whether the load task should be aborted due to preemption. | Called by ScaleSim's load task scheduler during runtime execution. |

**Fine-grained distance assignment.**  An agent may depend on multiple memory objects, and some of these objects can be shared across agents. For example, in prefix caching, agents often reuse overlapping KV prefixes organized in a tree structure. In such cases, a memory object may be associated with multiple agents that have different invocation distances. To support this, ScaleSim computes the invocation distance of a memory object as the minimum invocation distance among all agents that currently reference it. As illustrated in Figure 16, the middle memory objective is shared by Agent 1 and Agent 2, with distances of 1 and 2, respectively. In this case, the system assigns the memory object a distance of $\min\{1, 2\} = 1$. This fine-grained distance assignment allows ScaleSim to reason about memory relevance at the object level, rather than at the agent level, enabling more precise eviction and prefetching decisions in shared-memory settings.

**Load task scheduler.**  To manage memory load tasks efficiently, ScaleSim maintains a centralized scheduler that coordinates all memory transfers from CPU to GPU. This scheduler enables global prioritization based on invocation distance and avoids contention across independently issued loading operations. Without a centralized mechanism, competing memory modules may trigger redundant or conflicting loads, leading to bandwidth waste and degraded responsiveness under memory pressure. The `DispatchLoadTasks` interface in Table 1 allows developers to submit prefetch tasks to the scheduler.

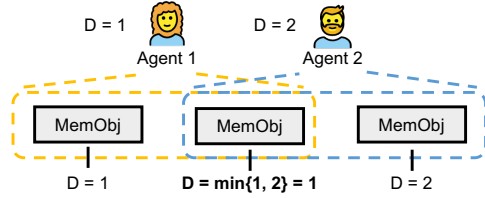

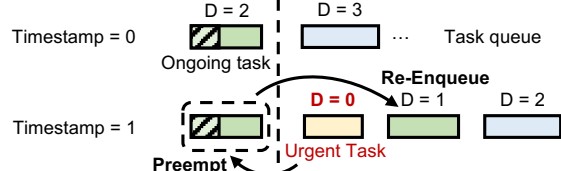

*Figure 16.* Example of memory object shared across agents. Invocation distance (D) is fine-grainedly assigned as the minimum among referencing agents.

*Figure 17.* Load task scheduler with preemption support. Urgent tasks can interrupt lower-priority loads and force re-enqueueing based on updated invocation distances.

**Preemption support.** In practice, invocation distance predictions are not always accurate. Unexpected events such as early termination of an action phase or a sudden change in agent behavior may cause an agent to reach the LLM stage earlier than anticipated. In such cases, a high-priority loading task may arrive while the system is already executing a lower-priority one. To handle this, ScaleSim supports preemption within its load task queue. As illustrated in Figure 17, at timestamp 0, the scheduler is executing a task with distance D = 2. At timestamp 1, a new urgent task with D = 0 arrives. The scheduler immediately interrupts the ongoing task, re-enqueues it back into the queue, and inserts the urgent task at the front. This design allows latency-critical agents to continue progressing even when prediction errors occur, while ensuring that the system falls back to the reactive baseline in the worst case. The preemption mechanism relies on the load interface's `GetPreemptionSignal` function handle, which developers can use to detect cancellation signals and safely terminate in-progress transfers.

## B. Detailed Related Work Comparison

ScaleSim targets memory management for general multi-agent simulation workloads, whereas most prior systems are designed for more specialized settings. Table 2 summarizes the key differences among representative systems.

*Table 2.* Comparison of ScaleSim with Existing Systems

| Systems | SGLang / vLLM | S-LoRA | FastLIBRA | InferCept | AI Metropolis | ScaleSim |
|---|---|---|---|---|---|---|
| Target Application | General LLM serving | Multi-LoRA LLM serving | Multi-LoRA LLM serving | Tool-intercepted generation | Generative-agent-style simulation | General multi-agent simulation |
| Optimized Memory Scope | General | LoRA adapters and KV cache | LoRA adapters and KV cache | KV cache only | None | General agent-specific memory |
| Loading Policy | Reactive | Prefetch adapters for queued requests | Application-unaware heuristic | Reactive | Reactive | Distance-guided prefetch |
| Eviction Policy | LRU | LRU | Application-unaware heuristic | Heuristic | LRU | Future reuse-aware eviction |
| Invocation Distance Capture | N/A | N/A | N/A | Limited | Limited | General abstraction with three categories |
| Sparsity Regime | N/A | N/A | N/A | Small | Large | Moderate to large |

Specifically, S-LoRA (Sheng et al., 2024) and FastLIBRA (Zhang et al., 2025a) optimize memory management in multi-LoRA LLM serving, but their designs do not consider multi-agent simulation workloads, ignoring application-aware optimization opportunities.

For AI Metropolis (Xie et al., 2024), it focuses on batch efficiency rather than memory management. Besides, the target application scope of ScaleSim is substantially broader than that of AI Metropolis. First, AI Metropolis models agent

dependencies primarily through spatial relationships, whereas ScaleSim supports a wider range of application semantics beyond spatial interactions. Second, the main optimization, out-of-order execution, in AI Metropolis cannot be applied to these applications. It assumes that action simulation does not affect subsequent LLM generations, which does not hold in our agent society (Piao et al., 2025) benchmarks where simulation affects subsequent generations. Third, AI Metropolis is evaluated only under extremely sparse settings, while ScaleSim continues to provide benefits under moderate sparsity.

For InferCept (Abhyankar et al., 2024), it estimates the next LLM invocation time based on profiled tool execution latency to determine whether to preserve or discard the KV cache. However, in general multi-agent simulation workloads, action simulation time can vary significantly across agents and across steps, and may further be affected by inter-agent interactions, which are not captured by InferCept's abstraction. In addition, InferCept relies on heuristics that require estimating absolute execution time to make eviction decisions. Such estimation is challenging in applications such as information diffusion and workflow-style simulations. In contrast, ScaleSim uses relative measures, such as hop count, to compare the urgency of different agents, enabling effective prefetching and eviction without accurate time prediction.

In summary, prior systems rely on specific assumptions or require accurate execution-time estimation, which limits their applicability to general multi-agent simulation. By introducing invocation distance as a unified abstraction, ScaleSim enables robust and effective memory optimization across a broad range of sparsity regimes and agent interaction patterns.

