# OpenReview forum: "ScaleSim: Serving Large-Scale Multi-Agent Simulation with Invocation Distance-Based Memory Management"
_ICML.cc/2026/Conference — ICML 2026 regular_

### Official Review · Reviewer_GfYv · 2026-03-09

**Soundness:** 3
**Presentation:** 4
**Significance:** 3
**Originality:** 4
**Overall Recommendation:** 5
**Confidence:** 4

**Summary:**

For multi-agent simulation systems supported by LLMs, I/O overhead substantially increases due to memory eviction and reloading if agents memory exceeds GPU capacity. Current LLM serving frameworks like vLLM and SGlang use generic memory management strategies, such as Least Recently Used (LRU) or usage-based heuristics, which may cause additional I/O overhead.


The authors made the following contributions:

They found two characteristics of multi-agent simulation workloads: 1) Sparse agent activation: the number of concurrent LLM requests at any given time is much smaller than the total number of agents; 2) Estimable agent invocation orders: they proposed the idea of invocation distance, which estimates the relative order in which agents will issue further LLM requests instead of the absolute time estimation. Further, they divide simulation applications into three types ( independent simulation, interaction-involved simulation, and predefined activation paths) and explain how to compute invocation distance for each.

Based on these findings, they proposed ScaleSim, an LLM serving system that efficiently manages GPU memory for large-scale multi-agent simulations. It included two key strategies to significantly reduce I/O overhead: 1) Proactive prefetching: it prefetches the memory of an agent in advance if the invocation distance is within a threshold; 2) Priority-based eviction: if the GPU memory is full, it evicts the agent memory with the largest invocation distance.

They implemented ScaleSim based on SGlang and conducted experiments using three representative applications in both single GPU and multi-GPU cases. The results showed that ScaleSim achieves 1.3-1.7x end-to-end speedup compared to baseline scenarios(SGlang without/with memory offloading).

**Compliance With Llm Reviewing Policy:**

Affirmed.

**Key Questions For Authors:**

Multi-agent simulation systems are typically needed because the behaviors and interactions among agents are difficult to predict using simple rules. I wonder whether the invocation order can truly be estimated in advance, especially for multi-agent simulation systems which involve intensive interactions. Could you provide more detailed examples showing how the invocation distance can be derived? It would be helpful if this process could be illustrated using the three representative applications in the evaluation section.

The paper does not discuss the overhead of calculating and maintaining the invocation distance. As the number of agents increases, the cost of computing this information may also increase and potentially offset the performance benefits. Also since agent interactions are dynamic, the invocation distances may need to be updated during the process. It would be helpful if you could explain and evaluate the overhead of calculating and maintaining invocation distance.

As mentioned in Strengths and Weaknesses - Soundness part, could you provide additional evaluation results for other two types of applications to show whether the same sparsity property still holds? Also in the evaluation section, it would be helpful if you could include other important metrics (TBT, throughput) to provide a more comprehensive evaluation.

**Limitations:**

The paper does not discuss the limitations in detail. The effectiveness of ScaleSim relies on two important assumptions: sparse agent activation and estimable agent invocation orders. This could potentially limit the applicability of ScaleSim to a broader range of applications.

**Strengths And Weaknesses:**

Strengths:

Soundness: Regarding the findings about multi-agent simulation workload characteristics, the authors provided experimental evidence for sparse agent activation as well as clear explanation about estimable agent invocation orders. Further, they conducted well-designed experiments to evaluate the performance gain from ScaleSim, which was well supported by the experiment results.

Presentation: The paper is well organized and easy to follow. Specifically, the examples (Figure 4 & Figure 7) used to explain the idea of ScaleSim are very illustrative.

Significance: The paper focused on optimizing the memory management for multi-agent simulation systems, which are important applications based on LLMs.

Originality: Different from current LLM serving frameworks like vLLM and SGlang, the paper proposed memory management strategies for multi-agent simulation systems with considering its specific workload characteristics, which largely reduces the I/O overhead. Additionally, different from previous optimizations which made strict assumptions about the workload pattern or predicted exact timestamps, the paper proposed the idea of invocation distance, increasing the applicability of their methods.


Weaknesses:
Soundness: They only showed sparse agent activation for independent simulation type (AgentSociety), while lack evidence for other two application types (interaction-involved simulation & predefined activation paths), which could show different patterns. Additionally, in the evaluation part, they only compared the end-to-end speedup and Time to First Token (TTFT). While other metrics like Time Between Tokens (TBT), throughput are also important during inference.

Presentation: None.

Significance: The effectiveness of ScaleSim relies on two important assumptions: sparse agent activation and estimable agent invocation orders. This could potentially limit the applicability of ScaleSim to a broader range of applications.

Originality: None.

---

> ### Author Rebuttal · Authors · 2026-03-30
>
> Thank you for your thoughtful feedback and positive assessment of our work.
>
> ## W1 & Q3. More Evaluation
>
> Thank you for the comments and questions.
>
> (a) Sparsity across different application types.
> We indeed include results for interaction-involved simulations and predefined activation paths in Figure 8 in the paper. These workloads also exhibit sparse agent activation, and ScaleSim consistently outperforms SGLang under these settings. We will revise the figure captions to make this observation more explicit and avoid potential misunderstanding.
>
> (b) Additional metrics.
> We agree that throughput is an important metric. We report throughput improvements over SGLang in the following table. ScaleSim improves throughput by reducing I/O stalls and increasing effective GPU utilization.
>
> **Table: Throughput (token/s) under AgentSociety Benchmark**
> |Simulated Agents|50|100|500|1000|
> |-|-|-|-|-|
> | SGLang  | 2563.5 | 3252.7 | 3926.8 | 3629.9 |
> | ScaleSim  | 2886.5 | 4091.9 | 5931.3 | 5778.8 |
>
> Regarding TBT, ScaleSim does not directly optimize it. As shown in Figure 13, the decoding latency remains comparable to the baseline. Our improvements mainly come from saving the memory loading time.
>
> ## W2. Application Range
>
> To demonstrate our wide application range, we further incorporate three more benchmarks that exhibit more complicated invocation patterns:
> (a) State-triggered simulation. Each agent maintains an internal state (e.g., "hunger level") that may interrupt its current action once a threshold is reached. We estimate the rate of state change based on recent observations to derive the invocation distance.
> (b) Dynamic social groups. In addition to physical distance constraints, agents interact only within their social groups, and these group memberships evolve over time. The invocation distance is thus defined based on dynamic social connectivity.
> \(c\) Stochastic environmental events. Beyond regular action execution, we introduce environmental events following a Poisson process, which may interrupt the simulation of a subset of agents. We incorporate the expected event trigger time when computing invocation distances.
>
> We include the corresponding experimental results in the following table, where ScaleSim consistently outperforms the baseline.
>
> **Table: ScaleSim Speedups over SGLang under Different Benchmarks**
> |Benchmarks\Simulated Agents|50|100|500|1000|
> |-|-|-|-|-|
> |State-triggered|1.13|1.26|1.49|1.59|
> |Dynamic social groups|1.08|1.16|1.16|1.32|
> |Stochastic events|1.11|1.18|1.27|1.29|
>
> We will add these results into the final version.
>
> ## Q1. Derivation of Invocation Distances
>
> Thank you for the question.
> In interaction-involved simulations, we use sandbox or game-engine-based environments [1,2,3] as illustrative examples. In such settings, agents are associated with positions in a virtual space. The interaction time between agents can be estimated based on their spatial distance and movement velocity, which are determined by their current states and actions.
>
>
> ## Q2. Overhead of Maintaining Invocation Distances
>
> The overhead of maintaining invocation distance is negligible, even at large scales.
> Invocation distance updates are executed asynchronously on CPUs and are lightweight. Importantly, they are not on the critical execution path. Our profiling shows that these updates can be fully overlapped with GPU execution, resulting in minimal overhead.
>
>
> ## Reference
>
> [1] Park, Joon Sung, et al. "Generative agents: Interactive simulacra of human behavior."
>
> [2] Ren, Jiawei, et al. "Simworld: An open-ended realistic simulator for autonomous agents in physical and social worlds."
>
> [3] https://simile.ai

---

> > ### Author Rebuttal · Reviewer_GfYv · 2026-04-04
> >
> > The authors have addressed the concerns and/or questions.

---

### Official Review · Reviewer_ti7G · 2026-03-12

**Soundness:** 3
**Presentation:** 2
**Significance:** 3
**Originality:** 3
**Overall Recommendation:** 4
**Confidence:** 5

**Summary:**

This paper, ScaleSim introduced a agentic offloading strategy for multi-agent simulation. Using distance of agentic invocation as the score for the strategy. This brings up to 1.74x speedup upon SGLang.

**Compliance With Llm Reviewing Policy:**

Affirmed.

**Final Justification:**

The rebuttal addressed my concern

**Key Questions For Authors:**

1. Why sparse is inherent in agentic execution? Is this related to long non-LLM execution? Do we have profiled results of LLM execution time and non-LLM execution time for each agentic simulation workflow?

2. At Figure 13 in paper, loading time increase with agent numberes. I do not think placing over 1000 agents in one cluster is a practical use case. The paper only shows the speed up under specific concurent number. Can authors provide a speed up upon sglang in different simulated agent numbers?

**Limitations:**

Discussed in strength and weaknesses.

**Strengths And Weaknesses:**

Strength:
1. Extensive results of TTFT and TPOT upon multiple evaluation workflows and environments.
2. Detailed anaylise and profiling of different agentic simulation workflow.

Weakness:
1. Some claim and assumption are contraint to several agentic workflow but not a univerisal observation(See questions for details).
2. Multiagent Simulation is mainly used for society simulation. Not a general usecase.

---

> ### Author Rebuttal · Authors · 2026-03-30
>
> Thank you for your constructive comments.
>
> ## W1 & Q1. Sparsity in Multi-Agent Simulation
>
> Thank you for your question.
> Yes, sparsity in multi-agent simulation is a result of the non-LLM execution stage. As shown in Figure 3 in the paper, at any given time, only a subset of agents are actively issuing LLM requests, while others are performing non-LLM execution, leading to inherent sparsity in agent activation.
>
> To quantify this effect, we measure the execution breakdown in the AgentSociety benchmark. The results show that LLM generation accounts for 4.13s, whereas non-LLM execution takes 14.3s. This imbalance further explains why only a small fraction of agents are concurrently invoking LLMs.
>
>
> ## W2. Importance and Generality of Multi-Agent Simulation
>
> Recent works [1,2,3] suggest that multi-agent simulation has become an increasingly important paradigm, with emerging applications beyond traditional social simulation, including gaming environments and reinforcement learning trajectory generation.
>
> As model capabilities continue to advance, we expect multi-agent systems to play a broader role in complex, large-scale task execution. This trend makes efficient system support for scalability and resource management increasingly critical.
>
> ## Q2. Co-locating Thousands of Agents
>
> Thank you for the question. Due to the sparsity of agent activation, naively scaling out to more GPU devices reduces effective batch sizes and leads to underutilization. In contrast, ScaleSim improves utilization by enabling efficient co-location of a large number of agents on the same instance through proactive memory management.
>
> Regarding scalability across different agent counts, we have evaluated performance under varying numbers of simulated agents. As shown in Figure 8, ScaleSim consistently achieves speedups over SGLang across a wide range of agent counts. We also include the detailed results in the following table.
>
> **Table: Speedups over SGLang**
> | Benchmarks\Simulated Agent Counts | 50 | 100 | 500 | 1000 |
> |-|-|-|-|-|
> | Generative Agents (Interaction-Involved) | 1.09 | 1.10 | 1.31 | 1.28 |
> | Information Diffusion (Predefined Activation Paths) | 1.26 | 1.55 | 1.75 | 1.58 |
>
> ## Reference
>
> [1] Park, Joon Sung, et al. "Generative agents: Interactive simulacra of human behavior."
>
> [2] Ren, Jiawei, et al. "Simworld: An open-ended realistic simulator for autonomous agents in physical and social worlds."
>
> [3] https://simile.ai

---

> > ### Author Rebuttal · Reviewer_ti7G · 2026-04-02
> >
> > Thank you for your response. I maintain my score.

---

### Official Review · Reviewer_9636 · 2026-03-12

**Soundness:** 2
**Presentation:** 3
**Significance:** 2
**Originality:** 3
**Overall Recommendation:** 3
**Confidence:** 4

**Summary:**

The paper presents ScaleSim, an LLM serving system optimized for large-scale multi-agent simulations. The authors identify two key properties of these workloads: sparse agent activation and estimable invocation orders. To exploit these, they introduce "invocation distance," a unified abstraction that predicts the relative order of future LLM requests. ScaleSim uses this distance to guide proactive prefetching and priority-based memory eviction. The system is implemented on top of SGLang and evaluated on three benchmarks, showing up to 1.74x speedup.

**Compliance With Llm Reviewing Policy:**

Affirmed.

**Key Questions For Authors:**

### **Key Questions For Authors***

1. **Handling Uncertainty**: How does ScaleSim perform in scenarios where agent activation is triggered by stochastic environment events rather than fixed action timers or physical proximity?
2. **Prediction Error Penalty**: Could you provide a quantitative analysis of the performance penalty when the "invocation distance" is frequently incorrect? Specifically, does the overhead of frequent preemption  eventually lead to performance worse than the reactive LRU baseline?


3. **Scheduler Scalability**: For simulations with tens of thousands of agents, does the centralized load scheduler or the frequency of `UpdateAgentDistance` calls  become a computational bottleneck?

**Limitations:**

The authors correctly identify that speedup is bounded by the proportion of time spent in the decode phase. However, they do not adequately address the limitations of their spatial interaction assumptions in more complex social or logical agent environments.

**Strengths And Weaknesses:**

#### **Strengths**

* **Presentation**: The paper is exceptionally well-written and structured . The analysis of I/O bottlenecks and the inefficiency of reactive memory management in existing systems is clear and compelling.


* **Originality**: The "invocation distance" abstraction is a clever way to bridge the semantic gap between high-level agent logic and low-level system resource management.


* **Soundness**: Within its defined scope, the technical implementation—including the load task scheduler and preemption support—is logically sound and well-integrated


#### **Weaknesses**

* **Simplistic Benchmarks**: The three categories used for evaluation (Independent, Interaction-Involved, and Predefined Paths) represent highly deterministic and simplified behaviors. For instance, "independent simulation" assumes agents follow repetitive, predictable cycles. This makes the "invocation distance" trivial to calculate and lacks the complexity found in real-world multi-agent systems.


* **Naive Interaction Model**: The spatial interaction model relies on a simple formula where distance is physical distance divided by velocity. While this works for basic crowd simulations, it ignores the vast majority of agent interactions driven by semantic logic, social hierarchy, or environmental triggers that are not physically bound.


* **Lack of Stochastic/Dynamic Testing**: The paper fails to evaluate how the system handles stochastic or competitive environments (e.g., game theory agents or complex economic simulations) where LLM calls may be bursty and unpredictable.

---

> ### Author Rebuttal · Authors · 2026-03-30
>
> Thank you for your detailed and constructive feedback.
>
> ## W1. Simplistic Benchmarks
>
> Thank you for the comments.
> While the evaluated benchmarks may appear simple, they capture key structural properties commonly observed in real-world multi-agent simulations, including sparsity and order-predictability of agent activations [1,2,3]. These patterns are prevalent in many practical systems where agent behaviors follow structured or semi-structured dynamics.
>
> To further address this concern, we have extended our evaluation with more complex and diverse benchmarks that exhibit less deterministic and more dynamic invocation patterns:
> (a) State-triggered simulation. Each agent maintains an internal state (e.g., "hunger level") that may interrupt its current action once a threshold is reached. We estimate the rate of state change based on recent observations to derive the invocation distance.
> (b) Dynamic social groups. In addition to physical distance constraints, agents interact only within their social groups, and these group memberships evolve over time. The invocation distance is thus defined based on dynamic social connectivity.
> \(c\) Stochastic environmental events. Beyond regular action execution, we introduce environmental events following a Poisson process, which may interrupt the simulation of a subset of agents. We incorporate the expected event trigger time when computing invocation distances.
>
> We include the corresponding experimental results in the following table, where ScaleSim consistently outperforms the baseline.
>
> **Table: ScaleSim Speedups over SGLang under Different Benchmarks**
> |Benchmarks\Simulated Agents|50|100|500|1000|
> |-|-|-|-|-|
> |State-triggered|1.13|1.26|1.49|1.59|
> |Dynamic social groups|1.08|1.16|1.16|1.32|
> |Stochastic events|1.11|1.18|1.27|1.29|
>
> We will incorporate these results into the final version.
>
> ## W2. Naive Interaction Models
>
> Thank you for the comments. We would like to clarify that the physical distance model is only one instantiation of the invocation distance abstraction, rather than a limitation of the system itself. ScaleSim does not assume any specific form of interaction; instead, it relies on a user-defined metric that captures the relative likelihood or ordering of future LLM invocations.
>
> As demonstrated in the additional benchmarks above, invocation distance can naturally extend beyond physical proximity to incorporate semantic relationships, dynamic group structures, or internal agent states.
>
> Moreover, we note that physical-distance-based interaction models are still widely adopted in prior simulation systems [1,2,4,5], making them a reasonable benchmark.
>
> ## W3 & Q1. Stochastic Testing
>
> Thank you for the question.
> As described in the third benchmark in W1, we model environmental event using a Poisson process to capture stochastic behavior. The experimental results show that ScaleSim consistently outperforms the baseline in this setting, demonstrating robustness under uncertainty.
>
> ## Q2. Prediction Error
>
> To evaluate robustness under inaccurate predictions, we conduct experiments by injecting random agent triggers at varying levels, thereby introducing controlled prediction errors.
>
> We report performance comparisons with SGLang for 500-agent simulations under different prediction error rates in the following table. Although the speedup decreases as the prediction error increases, ScaleSim still outperforms SGLang even under high error rates.
>
> **Table: Speedups over SGLang with Different Error Rates**
> |Error Rates|<5\%|\~20\%|\~40\%|\~60\%|\~80\%|
> |-|-|-|-|-|-|
> |Speedup|1.38|1.30|1.27|1.10|1.05|
>
> ## Q3. Scheduler Scalability
>
> The scheduler does not become a bottleneck even at the scale of thousands of agents.
> This is because invocation distance updates are executed asynchronously on CPUs and are lightweight. Importantly, they are not on the critical execution path. Our profiling shows that these updates can be fully overlapped with GPU execution, resulting in negligible overhead.
>
>
> ## Reference
>
> [1] Park, Joon Sung, et al. "Generative agents: Interactive simulacra of human behavior."
>
> [2] Ren, Jiawei, et al. "Simworld: An open-ended realistic simulator for autonomous agents in physical and social worlds."
>
> [3] https://simile.ai
>
> [4] Wang, Zhilin, et al. "Humanoid agents: Platform for simulating human-like generative agents."
>
> [5] AL, Altera, et al. "Project sid: Many-agent simulations toward ai civilization."

---

> > ### Author Rebuttal · Reviewer_9636 · 2026-04-02
> >
> > Thanks for your response. I maintain my scores.

---

### Official Review · Reviewer_qbbP · 2026-03-12

**Soundness:** 4
**Presentation:** 3
**Significance:** 3
**Originality:** 4
**Overall Recommendation:** 5
**Confidence:** 3

**Summary:**

The article presents ScaleSim, a system capable of optimizing the GPU memory
usage for large-scale multi-agent simulations by enabling proactive memory
prefetching and select the right evictions of inactive agents. Both key
features are based upon the computation and prediction of the next LLM
inference invocation. Experiments are conducted to evaluate the performance
of ScaleSim, thus achieving an up to 1.74x speedup against SGLang strategy
in three different configurations: independent simulations,
interaction-involved simulations and predefined activation paths simulations.

**Compliance With Llm Reviewing Policy:**

Affirmed.

**Final Justification:**

The rebuttal addressed my main concerns and reinforced my prior recommendation.

**Key Questions For Authors:**

1. Even though the concept of distance between agents is highly understandable
   in the use case of predefined activation paths, using hops as a countable
   metric, this is not so clear for interaction-involved simulations. Can you
   explain how physical distance is determined and what does velocity mean?
   Is this a metric which variates through time? May it depend on agents
   pairings? I may suppose that the velocity of an agent can differ according
   to which agent it can interact with, maybe because of the physical location
   of each agent.

2. According to the lack of a perspectives section, what do you plan for the
   following of ScaleSim? Will you implement it in some known LLM frameworks?
   Will you make it open-source?

3. Is there way for improvement? Other testbeds that can be used to show the
   benefits of the solution?

**Limitations:**

yes

**Strengths And Weaknesses:**

The article is well-written and easy to follow. The structure is clear
and almost every concept is defined at the right time. As the article
focuses on how the memory is managed during agents invocation, the state
of the art is correctly discussed and shows why the existing solutions
present drawbacks ScaleSim answers to. The given solution brings high
speedup against compared methods and shows its benefits following multiple
criteria such as the time-to-first-token comparison, the sensitivity to
sparsity and even a scalability study across model and GPU configurations.

The article also discusses the overhead of ScaleSim system by pointing out
it is negligible according to the obtained gain. However, a conclusion for
the ScaleSim project is missing, maybe giving a way to retrieve the
system and thus allowing its utilization or addressing some perspectives
in case there is still room for improvement. Also, the document outline is
missing from the introduction.

---

> ### Author Rebuttal · Authors · 2026-03-30
>
> Thank you for your positive assessment and insightful questions.
>
> ## W1 & Q2. Implementation and Plans
>
> Thank you for the question. We build ScaleSim on top of SGLang, and we plan to open-source the system upon acceptance. ScaleSim provides a unified interface that allows users to define customized invocation distance metrics, enabling the system to adapt to a wide range of application scenarios while benefiting from our memory optimization techniques.
> Furthermore, we implement several representative applications based on this abstraction, as shown in Sec4 in the paper.
>
> ## Q1. Interaction-Involved Simulation
>
> Thank you for the question.
> Physical distance is one representative instantiation of our invocation distance abstraction in interaction-involved simulations. In many sandbox or game-engine-based environments [1,2,3], agents are associated with positions in a virtual space. In such cases, the distance between two agents can be naturally defined based on their spatial coordinates, and their velocity corresponds to their movement speed determined by their current actions.
>
> Both distance and velocity are time-varying and context-dependent metrics. They can evolve over time as agents change states or behaviors. Moreover, the invocation distance can indeed depend on agent pairings. For example, agents without social connections or interaction rules can be assigned effectively infinite distance, indicating that interactions are unlikely.
>
> Importantly, ScaleSim does not assume a specific definition of distance. Instead, it only requires that the distance metric provides a predictive signal of future invocation. This abstraction allows users to define domain-specific distance functions beyond physical proximity.
>
>
> ## Q2. Improvement and Additional Testbeds
>
> Potential improvements:
> We agree that there is room for further improvement. In particular, when scaling to a larger number of agents across multiple GPUs, combining ScaleSim with data parallelism introduces new challenges in load balancing and scheduling. Designing more sophisticated partitioning and scheduling algorithms to achieve better cross-GPU balance is an important direction for future work.
>
> Additional testbeds:
> We have extended our evaluation with additional benchmarks that exhibit diverse invocation distance patterns:
> (a) State-triggered simulation. Each agent maintains an internal state (e.g., "hunger level") that may interrupt its current action once a threshold is reached. We estimate the rate of state change based on recent observations to derive the invocation distance.
> (b) Dynamic social groups. In addition to physical distance constraints, agents interact only within their social groups, and these group memberships evolve over time. The invocation distance is thus defined based on dynamic social connectivity.
> \(c\) Stochastic environmental events. Beyond regular action execution, we introduce environmental events following a Poisson process, which may interrupt the simulation of a subset of agents. We incorporate the expected event trigger time when computing invocation distances.
>
> We include the corresponding experimental results in the following table, where ScaleSim consistently outperforms the baseline.
>
> **Table: ScaleSim Speedups over SGLang under Different Benchmarks**
> |Benchmarks\Simulated Agents|50|100|500|1000|
> |-|-|-|-|-|
> |State-triggered|1.13|1.26|1.49|1.59|
> |Dynamic social groups|1.08|1.16|1.16|1.32|
> |Stochastic events|1.11|1.18|1.27|1.29|
>
> We will add these experiments to the paper in the next version.
>
> ## Reference
>
> [1] Park, Joon Sung, et al. "Generative agents: Interactive simulacra of human behavior."
>
> [2] Ren, Jiawei, et al. "Simworld: An open-ended realistic simulator for autonomous agents in physical and social worlds."
>
> [3] https://simile.ai

---

> > ### Author Rebuttal · Reviewer_qbbP · 2026-04-02
> >
> > Thanks for your response. You answered my concerns.

---

### Decision · Program_Chairs · 2026-04-30

**Decision:**

Accept (regular)

**Comment:**

This paper introduces ScaleSim, an LLM serving system tailored for large-scale multi-agent simulations. By identifying the sparse agent activation and estimable invocation order in these workloads, the authors propose a novel "invocation distance" abstraction to guide proactive memory prefetching and priority-based eviction. The system demonstrates up to a 1.74x speedup over SGLang.

The review committee found the paper to be exceptionally well-written, with a clear and compelling analysis of I/O bottlenecks in existing frameworks. The originality of the "invocation distance" metric was highly praised by multiple reviewers as a clever method to bridge high-level agent logic with low-level resource management.

Initially, the scores were divided: two reviewers recommended Accept (5) , one recommended Weak Accept (4) , and one recommended Weak Reject (3). The primary concerns raised by the reviewers leaning toward borderline/reject centered around the simplicity and deterministic nature of the initial benchmarks, specifically questioning how the system would handle stochastic environments or complex semantic agent interactions. Additionally, there were requests for further evaluation metrics, such as system throughput.

During the rebuttal phase, the authors provided a highly effective and comprehensive response. They extended their evaluations to include more dynamic and less deterministic benchmarks, specifically testing state-triggered simulations, dynamic social groups, and stochastic environmental events driven by a Poisson process. The new data clearly demonstrated consistent speedups (up to 1.59x) across these complex scenarios. Furthermore, the authors provided the requested throughput metrics, showing significant improvements over the baseline (e.g., 5931.3 token/s vs 3926.8 token/s for 500 agents). They also committed to open-sourcing the system.

Crucially, following the rebuttal, all four reviewers explicitly selected the status "Fully resolved - My concerns have been adequately addressed". Although two reviewers opted to maintain their initial scores of 3 and 4, their acknowledgment that all critical concerns were resolved indicates that there are no remaining technical flaws or major limitations that would prevent publication.

Conclusion:The paper offers a highly original and impactful contribution to the optimization of large-scale LLM serving systems. The initial weaknesses regarding benchmark diversity were thoroughly and convincingly addressed during the rebuttal. Given the strong technical foundation, the impressive empirical results, and the successful resolution of all reviewer concerns, this paper represents a valuable addition to the conference. Therefore, I recommend Accept.